# Rapid short-term reorganization in the language network

**Gesa Hartwigsen[1,2]\*, Danilo Bzdok[3,4,5], Maren Klein[2], Max Wawrzyniak[2], Anika Stockert[2], Katrin Wrede[2], Joseph Classen[6], Dorothee Saur[2]**

[1]Department of Neuropsychology, Max Planck Institute for Human Cognitive and Brain Sciences Leipzig, Leipzig, Germany; [2]Language and Aphasia Laboratory, Department of Neurology, University of Leipzig, Leipzig, Germany; [3]Department of Psychiatry, Psychotherapy and Psychosomatics, RWTH Aachen, Aachen, Germany; [4]JARA-BRAIN, Jülich-Aachen Research Alliance, Germany; [5]Parietal team, INRIA, Neurospin, bat 145, CEA Saclay, Gif-sur-Yvette, France; [6]Human Cortical Physiology and Motor Control Laboratory, Department of Neurology, University of Leipzig, Leipzig, Germany

**Abstract** The adaptive potential of the language network to compensate for lesions remains elusive. We show that perturbation of a semantic region in the healthy brain induced suppression of activity in a large semantic network and upregulation of neighbouring phonological areas. After perturbation, the disrupted area increased its inhibitory influence on another semantic key node. The inhibitory influence predicted the individual delay in response speed, indicating that inhibition at remote nodes is functionally relevant. Individual disruption predicted the upregulation of semantic activity in phonological regions. In contrast, perturbation over a phonological region suppressed activity in the network and disrupted behaviour without inducing upregulation. The beneficial contribution of a neighbouring network might thus depend on the level of functional disruption and may be interpreted to reflect a differential compensatory potential of distinct language networks. These results might reveal generic mechanisms of plasticity in cognitive networks and inform models of language reorganization.

**\*For correspondence:**
hartwigsen@cbs.mpg.de

**Competing interests:** The authors declare that no competing interests exist.

## Introduction

The current knowledge of short- and long-term plasticity in the language network after stroke-induced aphasia is limited. For instance, it is still a matter of debate if the temporary recruitment of neighbouring, ipsilateral networks and / or homologous right-hemispheric regions after a lesion of one critical node in the left hemisphere is adaptive or maladaptive for stroke recovery (*Chrysikou and Hamilton, 2011*; *Hamilton et al., 2011*). To investigate the potential for rapid short-term reorganization and flexible redistribution of the functional weight in the language net-work, we combined controlled, focal virtual lesions in the healthy brain with effective connectivity analyses of neuroimaging data. We relied on an experimental task that required the rapid analysis of the meaning of sound patterns (i.e., semantic processing), which is crucial for efficient every-day communication in humans.

Previous studies associated semantic processing with increased neural activation of left-hemispheric temporal, inferior frontal and inferior parietal regions (*Binder et al., 2009*; *Vigneau et al., 2006*). In particular, left angular gyrus (AG) and anterior inferior frontal gyrus (aIFG) were identified as semantic key nodes by means of focal perturbations induced with transcranial magnetic stimulation (TMS) (*Devlin et al., 2003*; *Sliwinska et al., 2015*). Specifically, several TMS studies assigned the left aIFG a central role in semantic control processes (*Hoffman et al., 2010*; *Whitney et al.,*

**eLife digest** Taking part in a conversation requires us to extract meaning from a complex series of sounds by recognising words and phrases. We then need to decide on a response, and plan and execute the lip and tongue movements necessary to generate that response. Each of these processes – from analysing the meaning of words to producing speech – requires a distinct set of brain regions to work together. However, we know relatively little about how these regions interact with one another during specific language processes, or about what happens when key regions are damaged.

Hartwigsen et al. have now used a technique called TMS in healthy volunteers to temporarily disrupt the activity of individual brain regions with a role in language. TMS involves applying small magnetic fields to the scalp over a target brain area. The magnetic fields induce electrical currents in the underlying brain tissue and temporarily scramble its activity. Hartwigsen et al. examined how this affected the volunteers' performance on a language task, as well as the activity of other language areas.

Temporarily disrupting a single brain region involved in analysing the meaning of words reduced the activity of multiple other areas in the brain's language networks. The greater this reduction in activity, the more poorly the volunteers performed on a language task. However, not all brain regions showed reduced activity. Areas adjacent to the disrupted region, which normally process the sounds of words, increased their activity. This increase may have partially compensated for the effects of the TMS to help limit the language impairments.

These findings offer insights into how the brain reacts and adapts when key language areas are damaged, for example as a result of stroke. The next step is to determine the extent to which healthy brain tissue can take over from damaged regions, and whether we can target this process to improve recovery after stroke.

*2011*). Others demonstrated that a number of temporal regions are also crucial for different semantic aspects. These regions include the anterior temporal lobe (ATL) for core semantic representations and the posterior middle temporal gyrus (pMTG) as another key node for semantic control aspects (*Binney and Ralph, 2015*; *Davey et al., 2015*; *Jung and Lambon Ralph, 2016*; *Whitney et al., 2012*).

With respect to the interactions in the semantic network, it was shown that the contribution of the left aIFG to semantic decisions crucially depends on the functional integrity of the left AG (*Hartwigsen et al., 2016*). Hence, left AG was able to compensate for a focal perturbation of aIFG during semantic decisions unless it was additionally perturbed with TMS, indicating a strong interaction between both regions. In contrast, TMS over neighbouring supramarginal gyrus (SMG) selectively delayed phonological decisions (i.e., decisions on the sound of words) but not semantic decisions.

While the above-cited behavioural TMS studies clearly demonstrate, besides temporal regions, a critical contribution of AG and aIFG to semantic processing, they do not allow for any conclusions on the neural underpinnings of the observed flexible redistribution in the semantic network after a focal perturbation of the AG. Moreover, the precise role of the AG in semantic processing remains elusive. For instance, this region showed increased task-related activity for words as compared to non-words or concrete relative to abstract words (*Binder et al., 2009*; *Seghier, 2013*), but deactivation for other semantic tasks, with the amount of deactivation being correlated with increased task-difficulty (*Humphreys et al., 2015*). Besides, the AG also contributes to other functions outside the core language domain like episodic memory and social cognitive tasks (*Bzdok et al., 2016*). Hence, it is unclear whether the task-related upregulation of left AG during word processing is necessary for semantic computations or rather reflects more general processes related to the level of task difficulty (*Graves et al., 2017*; *Ralph et al., 2017*).

To elucidate the brain's ability for adaptive short-term plasticity in response to a controlled virtual lesion and the contribution of left AG to semantic processing, we applied focal continuous theta-burst stimulation (cTBS) over AG or neighbouring SMG prior to neuroimaging (*Figure 1*). The

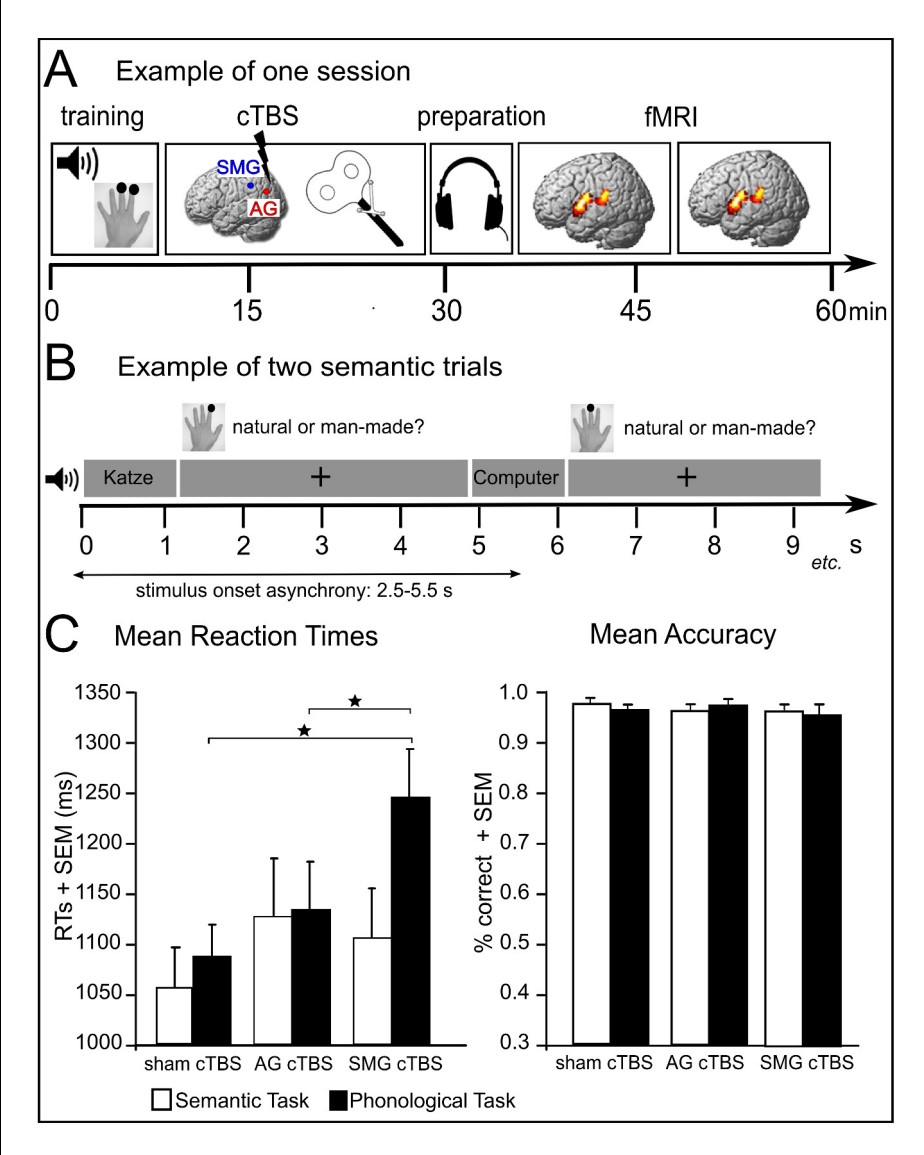

**Figure 1.** Experimental design and behavioural results. (**A**) Subjects received effective or sham cTBS either over supramarginal gyrus (SMG) or angular gyrus (AG) in different sessions. Thereafter, they performed semantic and phonological tasks in two fMRI runs. (**B**) Tasks were divided into 10 miniblocks per task and run, each consisting of 6 stimuli (e.g. 'Katze' ('*cat*')) with varying stimulus onset asynchrony. min=minutes; s=seconds. (**C**) Effects of cTBS over AG and SMG on reaction times (RTs) and accuracy. *p<0.05; SEM= standard error of the mean.

inclusion of a phonological task allowed us to test for the task specificity of the perturbation effect. cTBS was applied in healthy subjects to investigate immediate and transient perturbation effects. The lasting cTBS-induced suppression of neuronal excitability should give rise to an acute adaptive reorganization within the non-affected functional nodes of the network to compensate for the cTBS-induced suppression of neuronal activity (*Siebner and Rothwell, 2003*).

We expected to find a functional double dissociation of cTBS over AG vs. SMG on semantic vs. phonological decisions. Specifically, cTBS over AG should selectively inhibit task-related semantic activity at the targeted left AG, which might in turn lead to an upregulation of other important nodes in the semantic network (e.g. left aIFG, pMTG or ATL). Since our previous behavioural study (*Hartwigsen et al., 2016*) did not find a significant modulation of semantic response speed with a unifocal perturbation of AG, we did not expect any significant disruption of mean semantic response

speed after cTBS of AG. However, in accordance with the previous study, cTBS of SMG should delay phonological decisions, which might be reflected in the suppression of task-related activity in the phonological network.

As an alternative hypothesis, cTBS might also affect neural activity on a larger network level. Indeed, combined TMS-fMRI studies often reveal widespread stimulation effects, involving neighbouring and distant interconnected brain regions (*Bestmann et al., 2003*). If this were the case, then we would expect that cTBS over AG should decrease neural activity not only at the stimulated site itself, but in a larger semantic network.

As cognitive processes are mediated by the dynamic interactions among relevant areas rather than by isolated regions, effective connectivity analyses that capture perturbation effects on the causal network level should be more closely related to the neurobiological mechanisms by which a (virtual) lesion changes a cognitive function (*Hartwigsen et al., 2015*). Such network effects might also mediate possible cTBS-induced changes on the behavioural level.

## Results

### Differential effects of cTBS over AG and SMG on task-specific response speed

During fMRI, subjects performed semantic ('natural or manmade?') and phonological tasks ('two or three syllables?') on auditorily presented words. Response speed for correct trials was investigated with a repeated measures ANOVA including the within-subject factors task (semantic vs. phonological decisions) and cTBS (AG vs. SMG vs. sham). Overall, phonological response speed was significantly longer than semantic response speed (main effect of task: $F_{1,14}$ = 13.69, p=0.002; $\eta^2$= 0.56). However, this effect significantly interacted with cTBS (AG vs. SMG vs. sham), indicating that phonological and semantic decisions were differentially affected by cTBS ($F_{2,28}$ = 5.36, p=0.011; $\eta^2$= 0.28; *Figure 1C*). Two-tailed post-hoc paired t-tests revealed that cTBS over SMG significantly delayed phonological decisions when compared with sham cTBS ($t_{14}$ = 2.76, p=0.01; d = 0.73) or cTBS over AG ($t_{14}$ = 3.20, p=0.006; d = 0.81). In contrast, for semantic decisions, there was a trend towards increased response latencies after cTBS of AG relative to sham ($t_{14}$ = 2.15; p=0.05; d = 0.34) but not SMG cTBS (p=0.14; d = 0.13). Task accuracy was not affected by cTBS (all p>0.05).

### cTBS over AG decreases task-related activity in the semantic network and increases semantic activity in the phonological network

We first investigated the effects of cTBS over AG on semantic decisions. Relative to cTBS of neighbouring SMG, cTBS of AG significantly decreased task-related semantic activity not only at the stimulated area, but in a large network previously associated with semantic processing, including left AG (x,y,z = −42,–67, 28; T = 5.21); aIFG (x,y,z = −48, 41,–14; T = 6.64; x,y,z = −57, 26, 19; T = 6.48) and left posterior middle temporal gyrus (pMTG: x,y,z = −60,–43, −2; T = 5.55) (*Figure 2A*). Similar results for left AG and aIFG were obtained when contrasting cTBS of AG with sham cTBS (*Figure 2—figure supplement 1A* and *Table 1*). There was no significant difference in task-related activity between cTBS of SMG and sham cTBS during semantic decisions. This demonstrates the anatomical specificity of the perturbation effect.

A direct comparison of areas showing stronger activation increases after cTBS over left AG vs. SMG or sham cTBS during semantic decisions revealed an upregulation of several regions of the phonological network. Specifically, after cTBS over left AG vs. SMG, increased task-related neural activity was found in bilateral supramarginal gyrus (SMG, x,y,z = −45,–40, 46; T = 5.30; x,y,z = 42,,–44, 42; T = 5.00) and left posterior IFG/ventral premotor cortex (pIFG: x,y,z= −54, 5, 19; T = 5.28; *Figure 2B*). Similar results were found after cTBS over AG vs. sham cTBS (*Figure 2—figure supplement 1B* and *Table 1*). This upregulation presumably helped to restore task processing. The observed task-specific effects of cTBS over AG on semantic decisions are summarized in *Figure 3A*. Notably, the individual suppression of task-related activity in AG after cTBS over AG relative to sham cTBS predicted the individual upregulation of semantic activity in SMG (regression, normalized to sham cTBS, $R^2$ = 0.37, ß = −0.61, t = 2.79, p=0.016, two-tailed; *Figure 3B*).

Constrained probabilistic fiber tracking of diffusion tensor imaging data further revealed that the strong remote inhibitory effects of AG cTBS in the semantic network were most probably mediated

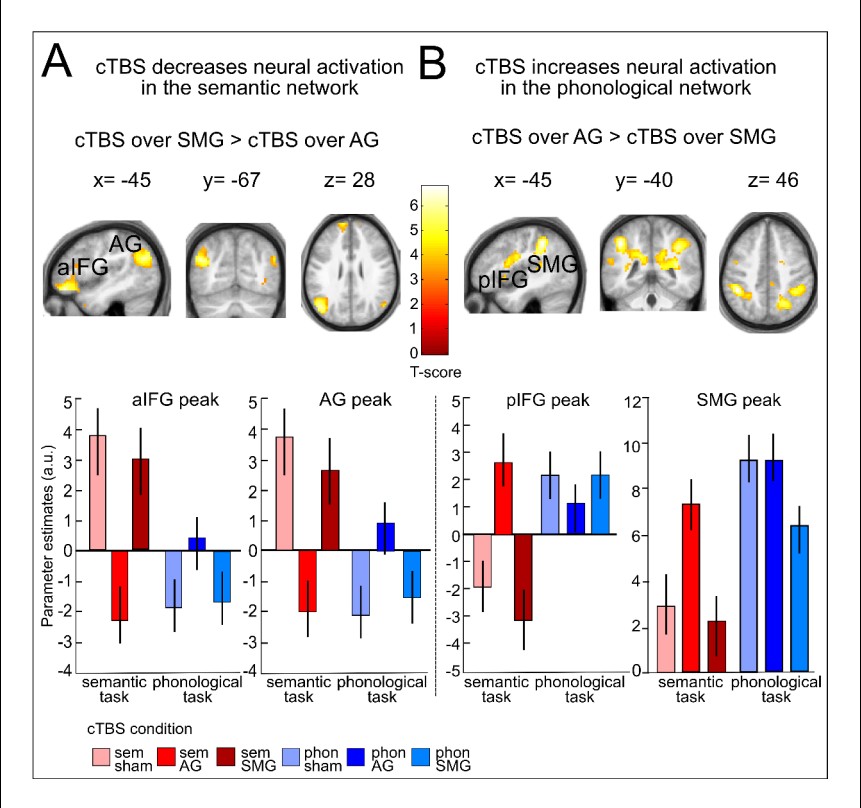

**Figure 2.** Effects of cTBS on semantic decisions. (**A**) Relative to cTBS over SMG, cTBS of AG significantly decreased neural activity not only at the stimulated area, but in a larger network including AG and aIFG. (**B**) Relative to cTBS of SMG, cTBS of AG significantly increased neural activity in phonological regions, including the bilateral SMG and pIFG. Lower panels display the respective parameter estimates (arbitrary units) for the different cTBS conditions that were extracted at the respective mean peak coordinates from the effect of interest for each task condition against rest. p<0.001 for display reasons. Sem=semantic, phon=phonological task.

The following figure supplements are available for figure 2:

**Figure supplement 1.** Effects of cTBS on semantic decisions.

**Figure supplement 2.** Task-related activity changes after sham cTBS (baseline).

by long-distance cortico-cortical association tracts. Specifically, AG and aIFG were connected via a ventral pathway constituted by the middle longitudinal fasciculus and the extreme capsule (*Saur et al., 2008*), also termed as inferior fronto-occipital fascicle (*Duffau et al., 2005*) (*Figure 3C*).

## cTBS over SMG decreases task-related activity for phonological decisions

For phonological decisions, we found a significant inhibition in the phonological network after cTBS over SMG (*Figure 4* and *Figure 4—figure supplement 1*). Specifically, when contrasted with cTBS over AG or sham cTBS, cTBS over SMG decreased task-specific activity in the bilateral SMG / superior parietal cortex, bilateral frontal operculum / pIFG and right supplementary motor area (*Table 1*). We did not find any upregulation of task-related activity during phonological processing after cTBS of SMG vs. cTBS of AG or sham cTBS, even after reducing the threshold. There were no significant differences in task-related activity between cTBS of AG and sham cTBS during phonological decisions.

**Table 1.** Changes in task-specific neural activation patterns after cTBS

| Region | Side | MNI coordinates X, Y, Z (in mm) | | | T | Cluster size |
|---|---|---|---|---|---|---|
| *Semantic judgements: SMG > AG cTBS* | | | | | | |
| inferior frontal gyrus (pars orbitalis) | L | −48 | 41 | −14 | 6.64 | 115 |
| inferior frontal gyrus (pars triangularis) | L | −57 | 26 | 10 | 6.48 | 95 |
| superior frontal gyrus | L | −9 | 44 | 43 | 5.71 | 92 |
| posterior middle temporal gyrus | L | −60 | −43 | −2 | 5.55 | 255 |
| angular gyrus | L | −42 | −67 | 28 | 5.21 | 240 |
| *Semantic judgements: sham > AG cTBS* | | | | | | |
| cerebellum | R | 24 | −85 | −38 | 5.21 | 117 |
| inferior frontal gyrus (pars orbitalis) | L | −45 | 38 | −14 | 5.12 | 45 |
| angular gyrus | L | −42 | −64 | 25 | 5.05 | 55 |
| *Semantic judgements: AG > SMG cTBS* | | | | | | |
| supramarginal gyrus | L | −45 | −40 | 46 | 5.30 | 215 |
| inferior frontal gyrus (pars opercularis) | L | −54 | 5 | 19 | 5.28 | 99 |
| supramarginal gyrus | R | 42 | −44 | 42 | 5.00 | 78 |
| *Semantic judgements: AG > sham cTBS* | | | | | | |
| supramarginal gyrus | L | −42 | −43 | 43 | 5.41 | 225 |
| supramarginal gyrus | R | 44 | −44 | 43 | 5.31 | 118 |
| inferior frontal gyrus (pars opercularis) | L | −57 | 8 | 16 | 5.28 | 103 |
| planum temporale | L | −57 | −40 | 19 | 5.01 | 35 |
| *Phonological judgements: AG > SMG cTBS* | | | | | | |
| supramarginal gyrus / superior parietal lobe | L | −45 | −41 | 42 | 5.24 | 222 |
| supramarginal gyrus / superior parietal lobe | R | 36 | −40 | 40 | 5.23 | 121 |
| frontal operculum / posterior inferior frontal gyrus | L | −55 | 10 | 4 | 4.97 | 87 |
| frontal operculum / posterior inferior frontal gyrus | R | 57 | 11 | 4 | 4.95 | 69 |
| supplementary motor area | R | 0 | 5 | 55 | 4.91 | 92 |
| *Phonological judgements: sham > SMG cTBS* | | | | | | |
| supramarginal gyrus / superior parietal lobe | L | −42 | −40 | 46 | 6.10 | 334 |
| frontal operculum / posterior inferior frontal gyrus | L | −51 | 8 | −2 | 5.91 | 97 |
| supramarginal gyrus / superior parietal lobe | R | 36 | −39 | 42 | 5.88 | 169 |
| frontal operculum / posterior inferior frontal gyrus | R | 57 | 8 | 7 | 5.44 | 82 |
| supplementary motor area | R | 0 | 7 | 55 | 5.29 | 101 |
| middle frontal gyrus | L | −33 | 41 | 25 | 5.12 | 57 |

thresholded at $p < 0.05$; FWE-corrected at the peak level, cluster extent >20 voxels.

## cTBS of AG increases the inhibitory influence of AG on aIFG during semantic decisions

We used dynamic causal modelling (DCM) to further explore how the inhibitory influence of cTBS over AG changed computations in the semantic network. DCMs were based on the main peaks obtained from the second-level fMRI analyses (i.e., left AG, aIFG and SMG). Among the 63 models tested (*Figure 5—figure supplement 1*), variational Bayesian model selection identified the model with driving input to SMG and modulation of the inhibitory connection from AG to aIFG as winning model. *Figure 5A* shows the winning model with the mean parameter estimates that were significantly different from zero (*Table 2*). These parameters included the intrinsic connection from AG to SMG (regardless of cTBS site, mean: 0.03, T = 3.27; p<0.006) and the modulation of the connection

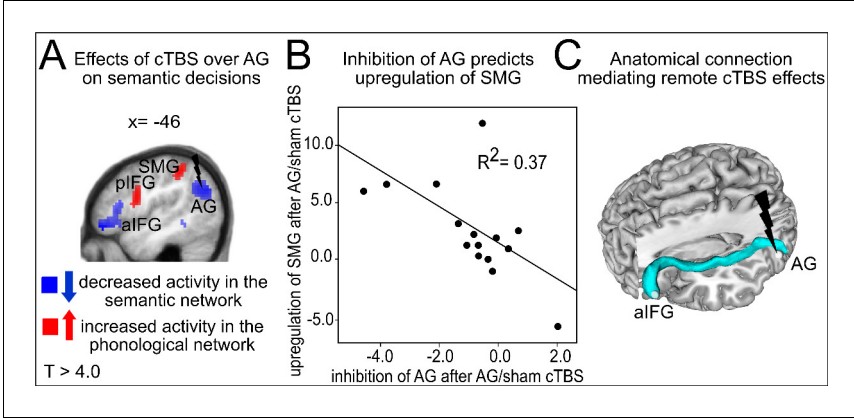

**Figure 3.** Semantic network effects. (**A**) Illustration of the strong cTBS-induced suppression in the semantic network (in blue) and the upregulation of the phonological network (in red). (**B**) The strength of the individual inhibition of left AG after cTBS (effect sizes for AG/sham cTBS received from the effect of interest at x,y,z= −42,−64, 25) predicted the upregulation of left SMG (effect sizes for AG/sham cTBS extracted from the effect of interest at x,y,z= −45,−40, 46). (**C**) Three-dimensional tractography rendering illustrating the underlying anatomical fiber connections mediating the remote effects of cTBS. AG and aIFG were most probably connected via the middle longitudinal fasciculus and extreme capsule.

from AG to aIFG by cTBS of AG (mean: −0.19, T = 3.67 p<0.003). Further parameters that did not survive a Bonferroni-Holm correction were the driving input to SMG (mean: 0.01, T = 2.24, p<0.046) and the intrinsic connection from AG to aIFG (mean: 0.03, T = 2.21, p<0.049). Our winning model had an exceedance probability of 79%, while all other models had probabilities < 10%. A family comparison pooled across different driving inputs identified the family with modulation of the inhibitory connection from AG to aIFG as winning family with an exceedance probability of 76%. The winning model indicates a weak positive intrinsic connection from AG to aIFG during semantic decisions that was reversed to a strong inhibitory influence after cTBS of AG. In contrast, cTBS of SMG did not significantly influence the connection from AG to aIFG (p=0.23). A direct comparison of the parameter estimates for cTBS of AG vs. SMG confirmed that the connection between AG and aIFG was

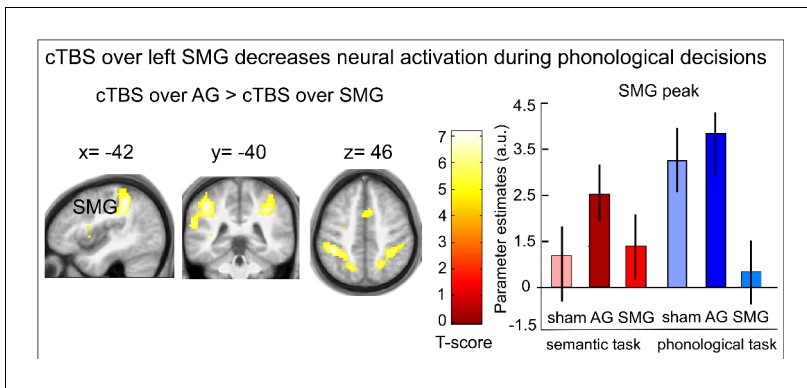

**Figure 4.** Effects of cTBS on phonological decisions. Relative to cTBS over AG, cTBS over SMG significantly decreased neural activity in bilateral supramarginal gyrus, with the strongest effect at the stimulation site. The right panel displays the parameter estimates (arbitrary units) for the different cTBS conditions that were extracted at the mean peak coordinates from the effect of interest for each task condition against rest. p<0.001 for display reasons.
The following figure supplement is available for figure 4:

**Figure supplement 1.** Inhibitory effects of cTBS on task-related neural activation during phonological decisions.

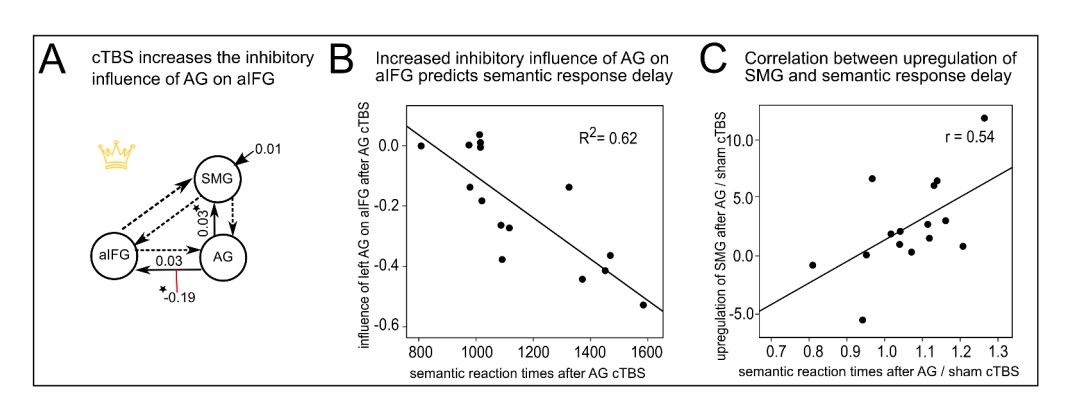

**Figure 5.** Effective connectivity in the semantic network. (**A**) The winning DCM model assumed modulation of the connection from left AG to aIFG by cTBS of left AG. Mean parameter estimates are given for the significant driving input to SMG, the facilitatory intrinsic connections from AG to aIFG and SMG (solid arrows) and the inhibitory modulation of the connection from AG to aIFG by cTBS over AG (red line), (*)survived a Bonferoni-Holm correction. (**B**) Regression analysis. The increase in the inhibitory influence of AG on aIFG after AG cTBS predicted the individual semantic response delay. (**C**) The degree of the individual upregulation of left SMG after cTBS of AG (effect sizes for AG/sham extracted from the effect of interest at x,y, z= −45,–40, 46) was significantly correlated with the delay in semantic response speed after AG/sham cTBS.

The following figure supplements are available for figure 5:

**Figure supplement 1.** Illustration of the different DCM-models.

**Figure supplement 2.** Regression analysis.

stronger modulated by cTBS of AG than SMG ($t_{14}$ = 2.92, p<0.026, paired t-test). Importantly, the individual increase in the inhibitory drive from AG to aIFG after cTBS of AG predicted the individual delay in the mean semantic response speed after cTBS (regression, $R^2$ = 0.62, ß = −0.79, t = 4.39, p=0.001, two-tailed; *Figure 5B*). These results show that the strong remote inhibition effects are behaviourally relevant.

**Table 2.** Mean parameter estimates of the winning model for semantic decisions

| Connection / parameter Right | Mean | SD | T | P |
|---|---|---|---|---|
| *Intrinsic connections* | | | | |
| AG→aIFG | 0.0295 | 0.0315 | 2.24 | 0.046 |
| AG→SMG | 0.0314 | 0.0140 | 3.27 | 0.006[*] |
| aIFG→AG | 0.1100 | 0.3965 | 1.07 | 0.30 |
| aIFG→SMG | 0.0529 | 0.1535 | 1.33 | 0.20 |
| SMG→AG | 0.2424 | 0.1642 | 2.10 | 0.056 |
| SMG→aIFG | 0.0413 | 0.1195 | 1.34 | 0.20 |
| *Modulation of connectivity from AG aIFG by cTBS* | | | | |
| cTBS of AG | −0.1900 | 0.1746 | −3.67 | 0.003[*] |
| cTBS of SMG | 0.0167 | 0.1471 | 1.27 | 0.23 |
| *Driving Input* | | | | |
| SMG | 0.0136 | 0.0106 | 2.21 | 0.049 |

[*]significant at p<0.05; two-tailed; corrected with a Bonferroni-Holm correction for multiple comparisons.

To account for the variability in the individual semantic response speed at baseline (i.e., after sham cTBS), we calculated another regression analysis with the cTBS-induced changes in the individual connectivity strength from AG to aIFG as predictor and the ratio of semantic response speed after AG cTBS vs. sham cTBS (i.e., baseline corrected) as dependent measure. Results were still significant after baseline correction ($R^2$ = 0.48; ß = −0.69; t = 3.79, p=0.006; two-tailed; see *Figure 5— figure supplement 2*).

### Strong individual disruption of the semantic network necessitates a stronger contribution of the SMG

We found a significant correlation between the individual upregulation of SMG after cTBS over AG and the cTBS-induced delay in mean semantic response speed (normalized to sham cTBS, r = 0.54, p=0.036, two-tailed; *Figure 5C*). This result indicates that those subjects who showed a strong disruption of the semantic network displayed a stronger upregulation of the neighbouring SMG. We speculate that this upregulation might have partially compensated for the cTBS-induced perturbation and enabled to maintain task processing.

### Dissociable parieto-frontal networks for semantic and phonological decisions

To explore the role of the observed parieto-frontal networks in semantic and phonological processing at baseline, we contrasted each task with rest after sham cTBS. Both tasks engaged widespread parieto-temporo-frontal networks of brain regions previously associated with both processes (*Devlin et al., 2003*; *Price et al., 1997*; *Vigneau et al., 2006*) (*Table 3* and *Figure 2—figure supplement 2*). Notably, we found increased task-related activity for (bilateral) AG and left aIFG during semantic decisions only, while phonological decisions increased task-specific activity in (bilateral) SMG and left pIFG (*Figure 2—figure supplement 2A,B*). A direct contrast of both tasks confirmed the role of AG and left aIFG (among other regions) as semantic areas (*Figure 2—figure supplement 2C*) and the contribution of SMG and left pIFG to phonological decisions (*Figure 2—figure supplement 2D*). This is well in line with previous fMRI studies that found similar regions for the respective contrasts of semantic and phonological decisions (*Devlin et al., 2003*; *Price et al., 1997*; *Vigneau et al., 2006*).

## Discussion

This study demonstrates functionally specific reorganization in the healthy semantic network after focal perturbation of a semantic key region. Our main novel finding was that cTBS over AG significantly decreased task-specific neural activity in a larger network of semantic regions, including the stimulated AG and left aIFG. Moreover, semantic suppression induced upregulation in neighbouring phonological regions that might have partially compensated for the disruptive effect on semantic decisions. In contrast, cTBS over SMG selectively affected phonological decisions, leading to decreased neural activity at the stimulated site and other phonological regions without any upregulation of the semantic network, as well as a significant delay in response speed. The observed functional-anatomical double dissociation during semantic vs. phonological decisions supports the notion that both processes are subserved by different networks (*Hartwigsen et al., 2016*) and suggests that these networks might differ in their potential to partially compensate for a focal disruption. The distinct contribution of AG and SMG to semantic and phonological decisions was further supported by their differential engagement in these processes after sham cTBS and by the respective differential task comparisons.

Notably, the task-specific perturbation effect in the semantic network was underpinned by a strong increase in the inhibitory influence from AG to aIFG, which predicted an individual delay in semantic performance after cTBS. In particular, individual response speed was prolonged as the inhibitory influence from left AG to aIFG increased. These findings suggest that a focal perturbation over a semantic key region can induce large-scale inhibitory effects on the network level. This provides new insight into the strong modulatory potential of cTBS. Probabilistic fiber tracking revealed that this effect was most probably mediated via anatomical long-distance connections that are part of the ventral association fibre system that was previously associated with semantic processing (*Bajada et al., 2015*).

**Table 3.** Changes in task-specific neural activation patterns after sham cTBS

| Region | Side | MNI coordinates X, Y, Z | | | T | Cluster size |
|---|---|---|---|---|---|---|
| *Semantic judgements > rest* | | | | | | |
| cerebellum (lobule VIIa) | R | 18 | −82 | −35 | 10.76 | 2981 |
| angular gyrus | L | −44 | −66 | 25 | 10.36 | 6144 |
| supplementary motor area | R | 5 | 14 | 52 | 10.21 | 354 |
| superior frontal gyrus | L | −12 | 38 | 46 | 9.47 | 2900 |
| inferior frontal gyrus (pars orbitalis) | L | −48 | 38 | −14 | 9.42 | 3200 |
| inferior frontal gyrus (pars triangularis) | L | −57 | 26 | 10 | 4.85 | subcluster |
| thalamus | L | −18 | −13 | 9 | 8.67 | 135 |
| postcentral gyrus | L | −57 | −19 | 25 | 6.33 | 68 |
| angular gyrus | R | 47 | −64 | 25 | 8.37 | 150 |
| middle temporal gyrus (posterior part) | L | −63 | −43 | −2 | 5.29 | 130 |
| precuneus | L | −3 | −52 | 16 | 4.87 | 1119 |
| *Phonological judgements > rest* | | | | | | |
| supramarginal gyrus / superior parietal lobe | L | −48 | −38 | 46 | 12.84 | 462 |
| superior parietal lobe | L | −27 | −57 | 44 | 8.67 | subcluster |
| supplementary motor area | R | 3 | 14 | 49 | 12.32 | 4936 |
| thalamus | L | −18 | −12 | 8 | 10.54 | 95 |
| thalamus | R | 15 | −13 | 13 | 10.93 | 98 |
| precentral gyrus / posterior inferior frontal gyrus (pars opercularis) | L | −45 | 5 | 25 | 8.76 | 5222 |
| middle / inferior frontal gyrus (pars triangularis) | L | −48 | 33 | 26 | 8.23 | subcluster |
| anterior insula | L | −30 | 20 | 7 | 10.21 | 326 |
| supramarginal gyrus / superior parietal lobe | R | 54 | −31 | 52 | 10.22 | 663 |
| precentral gyrus / primary motor cortex | R | 39 | −22 | 52 | 7.53 | subcluster |
| precentral gyrus | R | 36 | −19 | 67 | 7.37 | subcluster |
| cerebellum (lobule VI) | R | 30 | −64 | −29 | 10.12 | 2582 |
| cerebellum | R | 39 | −52 | −32 | 9.92 | subcluster |
| cerebellum | L | −18 | −52 | −23 | 9.14 | subcluster |
| inferior temporal gyrus | L | −51 | −52 | −14 | 6.53 | 154 |
| middle frontal gyrus | R | 45 | 35 | 19 | 8.67 | 345 |
| inferior frontal gyrus (pars opercularis) | R | 50 | 16 | 4 | 6.80 | subcluster |
| postcentral gyrus | L | −60 | −16 | 22 | 5.24 | 29 |
| *Semantic judgements > phonological judgements* | | | | | | |
| inferior frontal gyurs (pars triangularis) | L | −53 | 26 | 10 | 6.68 | 336 |
| angular gyrus | L | −45 | −67 | 28 | 6.21 | 348 |
| superior frontal gyrus | L | −9 | 59 | 28 | 5.88 | 292 |
| superior frontal gyrus | R | 9 | 41 | 49 | 4.95 | subcluster |
| angular gyrus | R | 54 | −64 | 25 | 4.92 | 123 |
| middle temporal gyrus (posterior part) | L | −63 | −40 | −2 | 4.90 | 139 |
| middle temporal gyrus (anterior part) | L | −54 | −2 | −23 | 4.88 | 28 |
| *Phonological judgements > semantic judgements* | | | | | | |
| inferior frontal gyrus (pars opercularis)/ frontal operculum | L | −51 | 8 | 4 | 7.21 | 1235 |
| supramarginal gyrus / superior parietal lobe | L | −42 | −37 | 40 | 6.84 | 158 |
| cerebellum (lobule VIIb) | R | 21 | −70 | −47 | 6.34 | 97 |

*Table 3 continued on next page*

*Table 3 continued*

| Region | Side | MNI coordinates X, Y, Z | | | T | Cluster size |
|---|---|---|---|---|---|---|
| *Semantic judgements > rest* | | | | | | |
| supplementary motor area | M | 3 | 5 | 61 | 6.23 | 383 |
| supramarginal gyrus / superior parietal lobe | R | 36 | −40 | 37 | 6.22 | 548 |
| middle frontal gyrus | L | −33 | 41 | 25 | 5.83 | 119 |
| inferior frontal gyrus (pars opercularis) | R | 54 | 11 | 7 | 5.67 | 242 |
| cerebellum (lobule VI) | R | 18 | −70 | −17 | 5.64 | 254 |
| middle frontal gyrus | R | 33 | 35 | 31 | 5.42 | 166 |
| superior temporal gyrus | L | −60 | −15 | 10 | 4.85 | 35 |

thresholded at p<0.05; FWE-corrected at the peak level, cluster extent >20 voxels.

We did not find a significant change in the mean semantic response speed after AG cTBS. However, a trend towards delayed semantic response speed with cTBS over AG indicated that this region contributes to semantic processing. We speculate that the relatively weak effect of AG cTBS on semantic decisions might be explained by the upregulation of the neighbouring phonological network that might have helped to maintain task processing. Indeed, the subject-specific degree of AG suppression after cTBS predicted the individual upregulation of neighbouring SMG. These results suggest that a topographically adjacent, yet functionally separate network may have the potential to partially support processing after disruption of a strategic key region in the language system. This interpretation is in line with our finding that the individual delay in semantic performance was correlated with the upregulation of the SMG after cTBS over AG. We conclude that a stronger individual disruption of the semantic network required a stronger contribution of the SMG to maintain task function. Together, our findings may show the inherent potential of the language system to flexibly recruit neighbouring regions, thus providing new insight in the dynamic regulation of intrahemispheric interactions in two functionally segregated processing streams. Specifically, the differential behavioural effects of cTBS over AG and SMG during semantic and phonological decisions suggest that lower-level resources like phonological working memory capacities (*Nixon et al., 2004*; *Romero et al., 2006*) might be recruited to partially contribute to higher-level semantic tasks to preserve task processing, but not the other way round. To the contrary, we observed a strong perturbation effect of SMG cTBS on task-related phonological activity and behaviour but no upregulation of semantic regions.

As an equally plausible alternative explanation, AG perturbation might have increased the overall task demands, thereby necessitating the contribution of more general executive control regions. Indeed, the upregulation of left SMG after cTBS of AG extended into the intraparietal sulcus (IPS) and superior parietal lobe (SPL) that were previously associated with executively demanding tasks (*Humphreys et al., 2015*) and are part of a multi-domain control system (*Duncan, 2010*; *Whitney et al., 2012*). Hence, AG perturbation might have increased the overall task demands, thereby requiring the contribution of these regions. In contrast, the strong perturbation effect on the phonological task after cTBS over SMG might be explained by the fact that (higher-level) semantic regions were not able to support task processing after disruption of the control network.

With respect to the role of the observed regions in semantic processing, both AG and aIFG were associated with executive control over semantic processing, although to a different degree (*Noonan et al., 2013*). Left AG corresponds to a high-level cross-modal convergence zone for concept retrieval and the integration of meaning and event representations and provides access to semantics (*Binder et al., 2009*). In particular, the ventral AG (overlapping with our stimulation site) was associated with automatic semantic retrieval, while a more dorsal subregion within the AG was assigned a role in controlled semantic retrieval (*Noonan et al., 2013*). aIFG, on the other hand, is an important node for top-down control during semantic processing and lexical retrieval (*Whitney et al., 2012*). Consequently, the observed network effect in our study might indicate that

disruption of AG induced a functional disconnection by impairing the transfer of stimulus related activity from AG to aIFG, thereby necessitating the contribution of more domain-general resources under semantic conditions with increased task difficulty. However, the precise role of the AG in semantic processing remains a conundrum. A recent meta-analysis showed that most semantic tasks deactivate the core AG and rather activate neighbouring intraparietal sulcus (among other regions) (*Humphreys et al., 2015*). On the other hand, left AG is clearly engaged by non-semantic domains such as memory and social cognition (*Bzdok et al., 2016*; *Humphreys et al., 2015*). Its precise contribution in semantic processing thus remains elusive. In our study, the semantic task might have particularly engaged semantic working memory processes that contributed to the observed positive AG activation (*Vigneau et al., 2006*).

We believe that our results are noteworthy for several reasons. First, they implicate that the effects of a focal perturbation are not restricted to the stimulated area itself but might rather affect a larger network (see *Bestmann et al., 2003*). Specifically, our data indicate that remote effects may occur in distant cognitive network nodes. Notably, the strong modulation of task-specific activity patterns in the absence of a significant disruption of the mean semantic response speed after AG cTBS indicates that behavioural measures alone might not be sufficient to map a cTBS-induced perturbation effect. Rather, some modulatory effects induced by non-invasive brain stimulation might only be captured on the neural network level and most likely originate from a modulation of the effective connectivity in a network. These results provide new insight into the neural mechanisms of the perturbation effect. Indeed, our winning model revealed a positive intrinsic connection between AG and aIFG, indicating that without cTBS, there might be a (weak) facilitatory connectivity between both regions that turned into a strong inhibition after cTBS. The individual magnitude of the induced inhibitory drive between AG and aIFG was able to predict the individual delay in semantic response speed after AG cTBS, demonstrating the functional relevance of this effect.

Secondly, the observed upregulation of a neighbouring parieto-frontal network for phonological processing might indicate some degree of degeneracy (*Price and Friston, 2002*) in the language network that might have partly enabled cross-modal compensation even after inhibition of a large part of the semantic network. This is compatible with the notion that cTBS may give rise to an acute adaptive reorganization within the non-targeted functional loops of another network to compensate for the cTBS-induced suppression of task-specific neuronal activity (*Hartwigsen and Siebner, 2012*).

The observed greater resilience of the semantic relative to the phonological network against the behavioural lesion effect was underpinned by a difference in the flexible recruitment of neighbouring networks that was selectively observed for the semantic task. These findings are in line with previous behavioural results that a single perturbation of one semantic node did not affect semantic decisions, while phonological decisions were already impaired after perturbation of one key region (*Hartwigsen et al., 2016*). The difference in the flexible recruitment of neighbouring networks for semantic and phonological processes likely reflects the engagement of higher- vs. lower-level processes in both tasks. Indeed, our optimal DCM identified the SMG as the most likely source of the driving input among the selected regions. This indicates that left AG receives information from the neighbouring SMG, which is compatible with the above-discussed role of the AG as a heteromodal integration area (*Seghier, 2013*). This might explain the absence of any upregulation of the semantic system during phonological processing after cTBS over SMG, as processing was already disrupted at a lower level. Interestingly, disruption of this lower-level region was not sufficient to significantly affect semantic task processing.

We initially hypothesized that the virtual lesion of the AG might lead to an upregulation of other semantic key nodes. However, this is not consistent with the observed strong remote effects of cTBS. It is worth noting that few imaging studies have investigated short-term plasticity on the neural-network level in the healthy language network to date (*Binney and Ralph, 2015*; *Hallam et al., 2016*; *Hartwigsen et al., 2013*; *Jung and Lambon Ralph, 2016*). Some of these studies (*Binney and Ralph, 2015*; *Hartwigsen et al., 2013*; *Jung and Lambon Ralph, 2016*) reported increased task-related activity of the right-hemispheric homologue after perturbation of a left-hemispheric key region for speech or language that was interpreted as adaptive short-term compensation. Another study (*Hallam et al., 2016*) found that perturbation of left IFG increased the activation of left pMTG during the processing of weak semantic associations, demonstrating that compensation can also occur between different semantic nodes within the same left hemispheric network. Moreover, *Jung and Lambon Ralph, 2016* showed that after cTBS over left ATL, the right ATL

increased its intrinsic facilitatory influence on left ATL, indicating a flexible, bilateral organization of the semantic system with a strong degree of adaptive plasticity.

In contrast to the above cited studies, we did not observe an upregulation of other left-hemispheric semantic areas or homologous right-hemispheric regions. When lowering the threshold, neural activity in right AG was also decreased after cTBS of left AG, indicating a strong cTBS-induced inhibition in the semantic network. This suggests that left AG is a strategic core region for semantic processes, a notion that is compatible with the key role of the AG in all aspects of semantic processing that require concept retrieval (*Binder et al., 2009*; *Seghier, 2013*), including efficient retrieval of semantic information (*Davey et al., 2015*).

Together, the previous findings and our results demonstrate the potential for a flexible recruitment after disruption of a semantic key node that may include regions within the same left-hemispheric network, as well as contralateral homologous regions or neighbouring regions from a different network. The exact recruitment may depend on the degree of the disruption and the specific task demands.

Our findings have implications for the interpretation of reorganization effects in the lesioned language network. The upregulation of neighbouring regions outside the core semantic network shows that aside from the specific contributions of a specialized network, neighbouring networks might also be beneficially recruited. In this context, our finding of a positive intrinsic connection between AG and neighbouring SMG during semantic processing argues against the possible alternative explanation of the observed upregulation of SMG in terms of disinhibition. Our findings are more compatible with the notion of an adaptive contribution of ipsilateral regions after a lesion of a strategic left-hemispheric area (*Heiss and Thiel, 2006*), rather than supporting the concept of maladaptive plasticity after a release from the inhibition of neighbouring regions.

Indeed, previous studies on post-stroke aphasia suggest that the upregulation of ipsi- vs. contralateral regions after stroke might depend on the site, size and extent of the lesion (*Chrysikou and Hamilton, 2011*). Models of language recovery proposed that for smaller lesions of critical left-hemispheric language nodes, neighbouring perilesional regions might be adaptively recruited to subserve language function while more severe impairments might stronger draw on recruitment of homotopic regions (*Hamilton et al., 2011*). Our results show that even a relatively focal perturbation can already inhibit neural activity in a large network. In our case, the recruitment of neighbouring areas seems to be related to a high level of adaptation (i.e., the absence of a strong behavioural effect), indicating a general flexibility of brain networks in terms of distributed processing.

In summary, our results shed important new light on the dynamic regulation of intrahemispheric interactions in the healthy human brain that are highly relevant for a variety of cognitive processes. This is of potential relevance for understanding language recovery after left hemisphere stroke, indicating that neighbouring networks might bear the inherent potential to partially support task processing after a strategic lesion of one key region. Our results help to unravel neural mechanisms of perturbation effects, indicating that cTBS not only influences neural activity in the stimulated region but also in remote areas. Although the observed effects were transient, they were sufficient to influence task-related activity, connectivity, and behaviour, as evidenced by the significant correlation between individual connectivity strength and delay in response speed. This demonstrates the value of combining TMS with effective connectivity analyses of fMRI data to map the neural consequences of a focal perturbation and might further inform models of language reorganization in post-stroke aphasia.

## Materials and methods

### Subjects

15 native, right-handed German speakers (seven females, age range: 20–30 years) with no history of neurological disorders or cTBS contraindications participated in our study. All subjects were included in the final analyses. Written informed consent was obtained before the experiment. The study was performed according to the guidelines of the Declaration of Helsinki and approved by the local ethics committee (Medical Faculty at the University of Leipzig).

## Experimental design

We used a two (*task: semantic vs. phonological decisions*) by three (*cTBS: effective cTBS of AG or SMG and sham cTBS*) factorial within-subject design (*Figure 1*). In three sessions (inter-session interval ≥7 days), we applied effective or sham cTBS over left AG or SMG prior to fMRI (*Figure 1A*). Each fMRI session was divided into two runs with a short break. During fMRI, subjects performed semantic ('natural or manmade?') and phonological tasks ('two or three syllables?') on auditorily presented words that were presented in short miniblocks of 6 items each (*Figure 1B*). Tasks were held constant within each miniblock and pseudorandomized across blocks such that no more than two blocks of the same task followed each other. We used similar items for both tasks that were pseudorandomized across conditions and counterbalanced across subjects (to the degree possible). Each session consisted of 10 miniblocks of each task, leading to a total presentation of 120 stimuli per session (60 for each task). Miniblocks were separated with 16 s rest periods. At the end of each rest block, task instruction was auditorily presented. Subjects were instructed to respond as quickly and accurately as possible by pressing a button on a response pad with their left middle or index finger. The stimuli were taken from a previous study (*Hartwigsen et al., 2010*). Only highly frequent, unambiguous nouns from the CELEX lexical database for German (Centre for Lexical Information, Max Planck Institute for Psycholinguistics, The Netherlands) were selected. In total, 60 two-syllable nouns and 60 three-syllable nouns were selected. All words represented natural or manmade items (i.e., 50% of the two-syllable words and 50% of the three-syllable words, respectively) and had a mean stimulus duration of 0.78 s (range = 0.74–0.87s).

## Magnetic resonance imaging

As a prerequisite for neuronavigated cTBS, all subjects first underwent structural MR imaging (Siemens Verio 3-Tesla scanner; Siemens, Erlangen, Germany). This included a high-resolution T1 weighted anatomical scan for each subject (MPRAGE; 170 slices, voxel size = $1 \times 1 \times 1.5$ mm, matrix = $240 \times 240$ pixel, TR = 1.3 s, TE = 3.46 ms). Functional imaging was performed using a gradient EPI sequence (repetition time [TR]/echo time [TE] = 2520/30 msec, flip angle = 90°, matrix = $64 \times 64$ pixel, voxel size = $3 \times 3 \times 3$ mm; field of view = 192 mm) with BOLD contrast for the acquisition of T2*-weighted images. A total of 370 volumes consisting of 38 slices was acquired continuously during each run in descending order.

Finally, we acquired axial whole brain diffusion weighted images with a double spin echo sequence for probabilistic fiber tracking (60 directions; *b*-value = 1000 s/mm$^2$; 88 slices; voxel size, $1.7 \times 1.7 \times 1.7$ mm, no gap; TR = 12.9 s; TE = 100 ms; field of view = $220 \times 220$ mm$^2$) plus seven volumes with no diffusion weighting (*b-value* = 0 s/mm$^2$) at the beginning of the sequence and interleaved after each block of 10 diffusion weighted images.

## Continuous theta-burst stimulation

We used neuronavigated cTBS (Brainsight; Rogue Research) based on coregistered individual T1-weighted MRI images to navigate the TMS coil and maintain its exact location and orientation throughout all sessions. Session order (AG, SMG or sham cTBS) was counterbalanced across subjects to the best possible degree. cTBS was performed using the mean Montreal Neurological Institute (MNI) coordinates for the left AG (x, y, z= −42,−66, 28 mm) and SMG (x, y, z= −45,−39, 45 mm) described in our previous study (*Hartwigsen et al., 2010*). Using these stereotactic coordinates, the individual stimulation sites were determined by calculating the inverse of the normalization transformation and transforming the coordinates from standard to individual space for each subject.

The TMS coil was positioned with the handle pointing lateral and perpendicular to the midline over left SMG or AG, with the second phase of the biphasic pulse inducing a lateral to medial current flow (*Hartwigsen et al., 2010*). We applied cTBS (600 stimuli at 50 Hz in trains of three stimuli at an inter-burst interval of 200 ms for 40 s) at a stimulation intensity of 80% of the individual active motor threshold (AMT). AMT was defined as the lowest stimulus intensity producing a motor evoked potential of 150–200 µV in the tonically active first dorsal interosseus muscle (20% of maximum contraction). A figure-of-eight-shaped coil (double 60 mm; coil type CB-60) connected to a MagPro X100 stimulator (MagVenture, Farum, Denmark) was used in all cTBS conditions. For sham cTBS, we used a specially designed figure-of-eight shaped placebo coil (MCF-P-B-70; outer diameter 7.5 cm) that creates a sound level identical to the CB-60 coil but provides an effective field reduction of

80%. Half of the subjects received sham cTBS over AG and SMG, respectively. The overall application of TMS pulses per sessions was well within safety limits.

## Data analyses fMRI data

Task-related changes in the blood oxygenation level-dependent signal were analyzed with SPM 8 and 12 (Wellcome Trust Centre for Neuroimaging; www.fil.ion.ucl.ac.uk/spm/) implemented in Matlab 8.1 (The Mathworks, Inc.,Natick, MA). Preprocessing of the fMRI data included slice timing, data realignment, coregistration of the individual T1-weighted and mean functional EPI images, segmentation, normalization into standard space (Montreal Neurological Institute (MNI) template) and smoothing with an isotropic 8 mm FWHM Gaussian kernel. Statistical analyses of the functional images were performed in two steps. The individual first level included regressors for each task condition, the instruction and the realignment parameters. All onsets in each regressor were convolved with a canonical hemodynamic response function as implemented in SPM8. Voxel-wise regression coefficients for all conditions were estimated using the least squares method within SPM8, and statistical parametric maps of the t statistic (SPM{t}) were generated from each condition (i.e., main effects of tasks under the different cTBS conditions). The data for the second stage of analysis comprised pooled parameter estimates for each of these contrasts across all participants in a random-effects analysis using a flexible factorial within-subject ANOVA design including a correction for non-sphericity. We used the Restricted Maximum Likelihood method in the SPM8 design specification for sphericity correction at the second-level inference. All comparisons were thresholded at a significance level of $p<0.05$, FWE-corrected at the peak level. The SPM anatomy toolbox (Version 1.7) and the WFU PickAtlas Tool (Version 2.4 Wake Forest University of School of Medicine) were used for anatomical localization of activation peaks.

## Dynamic causal modeling

To investigate the modulatory effects of cTBS in the semantic network, we employed effective connectivity analyses using dynamic causal modeling (DCM). DCM allows for investigation of the interaction of a predefined set of brain regions in different experimental contexts. The strength and direction of regional interactions are computed by comparing the observed regional blood oxygenation level-dependent (BOLD) responses with the BOLD responses predicted by a neurobiologically plausible model. The model describes how activity in, and interactions among, regional neuronal populations are modulated by external inputs (i.e., the experimental task conditions), as well as how the ensuing neuronal dynamics translate into the measured BOLD signal (*Stephan et al., 2010*). Three types of parameters are calculated: the direct influences of the external input or stimuli on regional activity (called *driving input*), the strength of the *intrinsic connections* between two regions in the absence of modulating experimental effects, and the changes in the intrinsic connectivity between regions induced by the experimental design (i.e., the modulatory effects of our cTBS conditions, including AG and SMG stimulation). The semantic network of interest included both cTBS sites (AG, SMG) and the aIFG. The model space for each task thus included 63 different models with full intrinsic connectivity (*Figure 5—figure supplement 1*). The driving input for our DCM analysis was set to either of the three regions of interest alone and to all possible combinations between these regions (*Figure 5—figure supplement 1A*) and the modulatory effects of cTBS over AG or SMG on the connections between these regions were specified for each subject (*Figure 5—figure supplement 1B*). The seed areas were derived from the comparison of SMG and AG cTBS for each task individually for each subject (mean coordinates for AG: x, y, z = −45,–67, 28; SMG: x, y, z= −45,–40, 46; aIFG: x, y, z = − −48, 41,–14). We first specified a new design matrix at the individual level that modelled the conditions of interest for the DCM analysis (i.e., one regressor modeling semantic judgements after cTBS over both AG and SMG, one regressor including only semantic judgements after cTBS of AG, and one regressor including only semantic judgements after cTBS of SMG). Subsequently, we extracted the first eigenvariate of the fMRI signal for our seed regions at the individual level at a liberal threshold of $p<0.01$ uncorrected within a sphere of 8 mm around the group coordinates derived from the effect of interest. These data were included in the model specification for our DCMs. For model selection across subjects, we used a random-effects Bayesian model selection procedure (*Stephan et al., 2010*) to identify the most likely model among the candidate models given the observed fMRI data. In this procedure, all models are tested against each other. The models'

probabilities are estimated, and inference is based on exceedance probabilities, reflecting the probability that this model is a better fit to the data than any other model of those tested. Finally, subject-specific estimates for the parameters of interest in the winning model (i.e., the driving input, the strength of the intrinsic connections between the seed regions, and the impact of the modulation on the relevant connections by cTBS of AG or SMG) were entered into Bonferroni-corrected two-sided one-sample t-tests to test differences from zero. We additionally performed a family-level inference based on model space partitioning (*Stephan et al., 2010*) to identify the most likely winning family. Accordingly, models were grouped based on the cTBS-induced modulation of the connection between regions. The family with the highest exceedance probability value again represents the model group that is most plausible given the data (*Stephan et al., 2010*).

### Probabilistic diffusion tensor Imaging-based fiber tracking

Diffusion weighted data sets were analyzed with the Leipzig Image Processing and Statistical Inference Algorithms (https://www.cbs.mpg.de/institute/software/lipsia) and the FMRIB's diffusion toolbox (FDT, Oxford Centre for Functional Magnetic Resonance Imaging of the Brain Diffusion Toolbox, Version 5.0. Preprocessing of the data included motion correction based on the 7 b0 images, global registration to the T1 anatomy and averaging of gradient directions resulting in an isotropic voxel resolution of 1 mm. Subsequently, a diffusion model was fitted to the preprocessed data with Bayesian estimation of diffusion parameters obtained using sampling techniques (*Behrens et al., 2007*). 8 mm spheres (seed masks) were created around the mean peak activation derived from the fMRI group contrasts of semantic decisions after AG cTBS vs. SMG cTBS (AG: x, y, z= −45,–67, 28; aIFG: x, y, z= −48, 41,–14) and transformed to the individual diffusion space.

Crossing fibre probabilistic tractography was performed based on each participant's probability distributions of voxel-wise principal diffusion directions applying region-to-region connectivity (*Behrens et al., 2007*). Region-to-region connectivity between the two seeds was assumed if repeatedly generated streamlines that started from one seed were passing through the other seed, that is, we picked those streamlines that passed through both ROIs in the sense of a constrained tractography. Finally, the estimated connectivity distributions were thresholded based on the total number of samples sent out that reached the respective other seed mask without being rejected. To remove spurious connections, individual tractography results were binarized at $1 \times 10^{-2}$ threshold percentage. These tracts were transformed into standard MNI space, and overlaid across subjects (threshold: $\geq$ 8 out of 15 subjects) using FSLView. Anatomical pathway definition was based on white matter atlases (Juelich Histological atlas and JHU White-Matter Tractography atlas distributed with FSL) and on the literature (*Saur et al., 2008*).

## Acknowledgements

GH was supported by the Deutsche Forschungsgemeinschaft (HA 6314/1–1). DS is supported by the James F McDonnell Foundation. None of the authors reported a conflict of interest.

## Additional information

### Funding

| Funder | Grant reference number | Author |
|---|---|---|
| Deutsche Forschungsgemeinschaft | HA-6314-1-1 | Gesa Hartwigsen |
| James F. MacDonnell Foundation | | Dorothee Saur |

The funders had no role in study design, data collection and interpretation, or the decision to submit the work for publication.

### Author contributions

GH, Conceptualization, Data curation, Formal analysis, Supervision, Funding acquisition, Investigation, Methodology, Writing—original draft, Project administration, Writing—review and editing; DB,

Data curation, Formal analysis, Methodology, Writing—review and editing; MK, MW, Data curation, Formal analysis; AS, Data curation, Formal analysis, Visualization; KW, Data curation; JC, DS, Conceptualization, Writing—review and editing

## Author ORCIDs

Gesa Hartwigsen, http://orcid.org/0000-0002-8084-1330

Danilo Bzdok, http://orcid.org/0000-0003-3466-6620

## Ethics

Human subjects: Written informed consent was obtained before the experiment. The study was performed according to the guidelines of the Declaration of Helsinki and approved by the local ethics committee (Medical Faculty at the University of Leipzig).

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
