## [Decision Letter]

Thank you for submitting your article "Rapid short-term reorganization in the language network" for consideration by *eLife*. Your article has been reviewed by two peer reviewers, and the evaluation has been overseen by a Reviewing Editor and Timothy Behrens as the Senior Editor. The following individuals involved in review of your submission have agreed to reveal their identity: Elizabeth Jefferies (Reviewer #2).

The reviewers have discussed the reviews with one another and the Reviewing Editor has drafted this decision to help you prepare a revised submission.

Summary:

The authors report that disruption of a semantic network node results in inhibition of remote semantic network nodes and releases activity on phonological networks. The extent of inhibition is correlated with semantic task performance. Disruption of phonetic networks however did not result in upregulation of semantic network activity and did disrupt phonological task performance. All reviewers agreed that this is an interesting paper which demonstrates an important dissociation of semantic versus phonological network activity in response to focal virtual brain lesions.

Essential revisions:

The main points raised by both reviewers focused on interpretation of the results and in particular whether strong claims for compensation could be supported by the data. Along these lines, the following specific points should be addressed:

1) We would recommend to not over-emphasize the notion of a compensatory activity increase. To convincingly and directly lend support to this claim, a double lesion study would need to be performed and demonstrate a decrease in semantic task performance after cTBS to both AG and SMG.

2) The authors are not clear about how to interpret the behavioral results of the cTBS over AG: Is there a significant reduction in reaction time or not? Depending on the answer, there are 3 possible interpretations: 1) If the answer is "yes" then it might be difficult to argue with a compensatory effect (because there is a behavioral deficit but one may speculate about a "partial" compensation). If the answer is "no", then the stimulated region 2) does not support the postulated function (which is unlikely but a possible interpretation or 3) its function has been compensated completely by a different region. Cases 1) and 3) would both require the experiment described above to draw a conclusion. The Discussion should maybe follow this line of argumentation.

3) The correlation in Figure 3 may provide some indirect support to the notion of " partial compensatory activation" but is definitely not causal proof.

4) The same holds the correlation in Figure 5. As per Figure 1, the variability of RT's during sham is similar to the variability during cTBS over AG. Does the correlation in Figure 5 still hold when corrected for baseline differences in RT's (i.e. adding RT during sham as covariate into the model)?

5) The compensation argument is difficult given the role of SMG in cognitive control: The study found that SMG stimulation disrupted the phonological judgements more than semantic judgements but this could be due to mismatched task difficulty – i.e., SMG is important for cognitive control. It falls within the saliency network. The authors want to interpret their findings as evidence that phonological processes can compensate for disruption to the semantic network, and not the other way round – and this is plausible. However, an equally plausible alternative could be that control regions are recruited to support semantic processing after disruption of the semantic network, while semantic regions cannot compensate for disruption to the control network. The discussion does later acknowledge the role of SMG in control – but this potentially contradicts the earlier argument that it is to do with an increased reliance on phonological resources. It would be better to acknowledge these two possibilities upfront, or to provide more evidence for a favoured interpretation, if the authors have one.

[Editors' note: further revisions were requested prior to acceptance, as described below.]

Thank you for resubmitting your work entitled "Rapid short-term reorganization in the language network" for further consideration at *eLife*. Your revised article has been favorably evaluated by Timothy Behrens (Senior Editor) and a Reviewing Editor.

The manuscript has been improved but there are some remaining issues that need to be addressed before acceptance, as outlined below:

The first essential revision encouraged the authors to tone down the arguments for compensation. In revisions, the authors have occasionally qualified claims with the addition of 'partial' but have not removed any instances of the word compensate/compensatory, which is still very prominent. These changes do not go far enough. The reviewers were concerned that the authors have not directly tested whether the observed overactivation is compensatory. Therefore, to describe it as 'compensatory overactivation' goes beyond the data. In general, this phenomenon should simply be described as overactivation or upregulation. It would be fine to mention that the overactivation may reflect compensation in some cases – but this is an interpretation, without data to directly back it up.

For example, in the Abstract, we suggest simply describing the overactivity as 'upregulation' and not as 'partial compensatory upregulation'. The concept of compensation could be mentioned in the final two sentences of the Abstract as a potential interpretation of the results found.

Similarly, at the end of the Introduction, "Specifically, cTBS over AG should selectively inhibit task-related semantic activity at the targeted left AG, which might in turn lead to a compensatory up-regulation of other important nodes in the semantic network (e.g. left aIFG, pMTG or ATL)." – remove 'compensatory' as you do not test whether or not it is compensatory.

In the Results, 'compensation' should only be used when it is clear that is an interpretation rather than what is measured. So, remove it from the second paragraph of the subsection “cTBS over AG decreases task-related activity in the semantic network and increases semantic activity in the phonological network”, but it's ok to use in the subsection “Strong individual disruption of the semantic network necessitates a stronger contribution of the SMG”.

In the Discussion, the new sentence "Moreover, semantic suppression induced upregulation in neighbouring phonological regions that might have partially compensated for the disruptive effect on semantic decisions" – is helpful. Whereas in the following sentence, "without any compensatory upregulation of the semantic network" – the word 'compensatory' should be removed.

For the remainder of the Discussion, the authors should carefully consider whether use of the term compensatory is appropriate.

---

## [Author Response]

Essential revisions:

The main points raised by both reviewers focused on interpretation of the results and in particular whether strong claims for compensation could be supported by the data. Along these lines, the following specific points should be addressed:

1) We would recommend to not over-emphasize the notion of a compensatory activity increase. To convincingly and directly lend support to this claim, a double lesion study would need to be performed and demonstrate a decrease in semantic task performance after cTBS to both AG and SMG.

We would like to thank the editor and reviewers for the overall positive evaluation of our manuscript. We recognize that the claims about the compensation of the semantic effect were too strong. In full agreement with the comment, these speculations should not be over-emphasized as we did not additionally target left SMG with TMS afterwards to provide causal evidence for this claim. In response to the reviewers’ suggestions, we have down-toned the respective paragraphs of the Discussion and rephrased our conclusions accordingly. We hope that we have answered all specific concerns as detailed below.

2) The authors are not clear about how to interpret the behavioral results of the cTBS over AG: Is there a significant reduction in reaction time or not? Depending on the answer, there are 3 possible interpretations: 1) If the answer is "yes" then it might be difficult to argue with a compensatory effect (because there is a behavioral deficit but one may speculate about a "partial" compensation). If the answer is "no", then the stimulated region 2) does not support the postulated function (which is unlikely but a possible interpretation or 3) its function has been compensated completely by a different region. Cases 1) and 3) would both require the experiment described above to draw a conclusion. The Discussion should maybe follow this line of argumentation.

We apologize for not being clearer in the initial version and would like to thank the reviewer for her / his suggestions on the interpretation of our findings as “partial compensation”. Indeed, we believe that the observed trend towards delayed response speed in the semantic task after cTBS over AG in combination with the upregulation of the neighbouring left SMG is most compatible with the notion of a partial compensation. As a consequence to the reviewer comment, we consistently use the term “partial compensation” throughout the Discussion in the revised manuscript.

To further clarify this issue, we have rewritten the respective passage of the Discussion as follows: “We did not find a significant change in the mean semantic response speed after AG cTBS. […] We conclude that a stronger individual disruption of the semantic network required a stronger contribution of the SMG to maintain task performance.”

3) The correlation in Figure 3 may provide some indirect support to the notion of " partial compensatory activation" but is definitely not causal proof.

We agree with the reviewer. As noted above, we have toned down the respective passage of the Discussion. The statement on the correlation now reads as follows: “Indeed, the subject-specific degree of AG suppression after cTBS predicted the individual upregulation of the neighbouring SMG. These results suggest that a topographically adjacent, yet functionally separate network may have the potential to partially compensate for a disruption of a strategic key region in the language system.”

4) The same holds the correlation in Figure 5. As per Figure 1, the variability of RT's during sham is similar to the variability during cTBS over AG. Does the correlation in Figure 5 still hold when corrected for baseline differences in RT's (i.e. adding RT during sham as covariate into the model)?

To address this useful reviewer comment, we have also toned down the conclusion on the prediction of the individual semantic response speed by the change in the influence from AG to aIFG after cTBS. The respective statement has been changed as follows:

“Notably, the task-specific perturbation effect in the semantic network was underpinned by a strong increase in the inhibitory influence from AG to aIFG, which predicted an individual delay in semantic performance after cTBS. In particular, individual response speed was prolonged as the inhibitory influence from left AG to aIFG increased. These findings suggest that a focal perturbation over a semantic key region can induce large-scale inhibitory effects on the network level.”

To account for the variability in the individual semantic response speed at baseline (i.e., after sham cTBS), we calculated another regression analysis with the cTBS-induced changes in the individual connectivity strength from AG to aIFG as predictor and the ratio of semantic response speed after AG cTBS vs. sham cTBS (i.e., baseline corrected) as dependent measure. Results were still significant after baseline correction (R^2^=0.48; ß= -0.69; t= 3.79, p=0.006, two-tailed; see Figure 5—figure supplement 2). This analysis has been included in the Results subsection “cTBS of AG increases the inhibitory influence of AG on aIFG during semantic decisions”.

5) The compensation argument is difficult given the role of SMG in cognitive control: The study found that SMG stimulation disrupted the phonological judgements more than semantic judgements but this could be due to mismatched task difficulty – i.e., SMG is important for cognitive control. It falls within the saliency network. The authors want to interpret their findings as evidence that phonological processes can compensate for disruption to the semantic network, and not the other way round – and this is plausible. However, an equally plausible alternative could be that control regions are recruited to support semantic processing after disruption of the semantic network, while semantic regions cannot compensate for disruption to the control network. The discussion does later acknowledge the role of SMG in control – but this potentially contradicts the earlier argument that it is to do with an increased reliance on phonological resources. It would be better to acknowledge these two possibilities upfront, or to provide more evidence for a favoured interpretation, if the authors have one.

We apologize for the identified source of confusion. We agree with the reviewer that both explanations are equally plausible. Indeed, our data is not suited to fully exclude either possibility. In response to the reviewer’s suggestion, we now discuss both explanations in the same passage.

Consequently, the respective part of the Discussion has been changed as follows:

“Specifically, the differential behavioural effects of cTBS over AG and SMG during semantic and phonological decisions suggest that lower-level resources like phonological working memory capacities (Nixon et al., 2004; Romero et al., 2006) might be recruited to partially contribute to higher-level semantic tasks to preserve task processing, but not the other way round. […] In contrast, the strong perturbation effect on the phonological task after cTBS over SMG might be explained by the fact that (higher-level) semantic regions were not able to compensate for disruption of the control network.”

[Editors' note: further revisions were requested prior to acceptance, as described below.]

The manuscript has been improved but there are some remaining issues that need to be addressed before acceptance, as outlined below:

The first essential revision encouraged the authors to tone down the arguments for compensation. In revisions, the authors have occasionally qualified claims with the addition of 'partial' but have not removed any instances of the word compensate/compensatory, which is still very prominent. These changes do not go far enough. The reviewers were concerned that the authors have not directly tested whether the observed overactivation is compensatory. Therefore, to describe it as 'compensatory overactivation' goes beyond the data. In general, this phenomenon should simply be described as overactivation or upregulation. It would be fine to mention that the overactivation may reflect compensation in some cases – but this is an interpretation, without data to directly back it up.

We agree with the editors and apologize for overestimating our findings. Following the editors’ suggestions, we have replaced “compensation” with “upregulation” in most cases. In some passages of the Discussion, we speculate that our data might reflect compensation in some cases. We hope that we have now sufficiently clarified that this is an interpretation (see reply 6 for details and examples). We also use the term “compensation” when citing other studies and their interpretation. Detailed changes are reported below.

For example, in the Abstract, we suggest simply describing the overactivity as 'upregulation' and not as 'partial compensatory upregulation'. The concept of compensation could be mentioned in the final two sentences of the Abstract as a potential interpretation of the results found.

Following the editors’ advice, we adjusted the respective parts of the Abstract:

“We show that perturbation of a semantic region in the healthy brain induced suppression of activity in a large semantic network and upregulation of neighbouring phonological areas.”

“In contrast, perturbation over a phonological region suppressed activity in the network and disrupted behaviour without inducing upregulation. The beneficial contribution of a neighbouring network might thus depend on the level of functional disruption and may be interpreted to reflect a differential compensatory potential of distinct language networks.”

Similarly, at the end of the Introduction, "Specifically, cTBS over AG should selectively inhibit task-related semantic activity at the targeted left AG, which might in turn lead to a compensatory up-regulation of other important nodes in the semantic network (e.g. left aIFG, pMTG or ATL)." – remove 'compensatory' as you do not test whether or not it is compensatory.

We have modified the respective passage of the Introduction accordingly:

“Specifically, cTBS over AG should selectively inhibit task-related semantic activity at the targeted left AG, which might in turn lead to an upregulation of other important nodes in the semantic network (e.g. left aIFG, pMTG or ATL).”

In the Results, 'compensation' should only be used when it is clear that is an interpretation rather than what is measured. So, remove it from the second paragraph of the subsection “cTBS over AG decreases task-related activity in the semantic network and increases semantic activity in the phonological network”, but it's ok to use in the subsection “Strong individual disruption of the semantic network necessitates a stronger contribution of the SMG”.

Following the editors’ suggestions, the respective passages of the Results now read as follows:

“Notably, the individual suppression of task-related activity in AG after cTBS over AG relative to sham cTBS predicted the individual upregulation of semantic activity in SMG.”

“We speculate that this upregulation might have partially compensated for the cTBS-induced perturbation and enabled to maintain task processing. “

In the Discussion, the new sentence "Moreover, semantic suppression induced upregulation in neighbouring phonological regions that might have partially compensated for the disruptive effect on semantic decisions" – is helpful. Whereas in the following sentence, "without any compensatory upregulation of the semantic network" – the word 'compensatory' should be removed.

We agree with the editors. Consequently, we have removed “compensatory” from the respective sentence:

“In contrast, cTBS over SMG selectively affected phonological decisions, leading to decreased neural activity at the stimulated site and other phonological regions without any upregulation of the semantic network, as well as a significant delay in response speed.”

*For the remainder of the Discussion, the authors should carefully consider whether use of the term compensatory is appropriate.*

We have carefully revised the Discussion to follow the editors’ advice. The respective passages on the upregulation of neighboring regions were changed as follows:

“The observed functional-anatomical double dissociation during semantic vs. phonological decisions supports the notion that both processes are subserved by different networks (Hartwigsen et al., 2016) and suggests that these networks might differ in their potential to partially compensate for a focal disruption.”

“We speculate that the relatively weak effect of AG cTBS on semantic decisions might be explained by the upregulation of the neighbouring phonological network that might have helped to maintain task processing. […] This interpretation is in line with our finding that the individual delay in semantic performance was correlated with the upregulation of the SMG after cTBS over AG.”

“To the contrary, we observed a strong perturbation effect of SMG cTBS on task-related phonological activity and behaviour but no upregulation of semantic regions. […] In contrast, the strong perturbation effect on the phonological task after cTBS over SMG might be explained by the fact that (higher-level) semantic regions were not able to support task processing after disruption of the control network.”

“Secondly, the observed upregulation of a neighbouring parieto-frontal network for phonological processing might indicate some degree of degeneracy (Price and Friston, 2002) in the language network that might have partly enabled cross-modal compensation […].”

“This might explain the absence of any upregulation of the semantic system during phonological processing after cTBS over SMG, as processing was already disrupted at a lower level. […] We initially hypothesized that the virtual lesion of the AG might lead to an upregulation of other semantic key nodes.”

“Together, the previous findings and our results demonstrate the potential for a flexible recruitment after disruption of a semantic key node […].”

“The upregulation of neighbouring regions outside the core semantic network shows that aside from the specific contributions of a specialized network, neighbouring networks might also be beneficially recruited.”

“This is of potential relevance for understanding language recovery after left hemisphere stroke, indicating that neighbouring networks might bear the inherent potential to partially support task processing after a strategic lesion of one key region.”

Figure 3 (Figure legend)

“Illustration of the strong cTBS-induced suppression in the semantic network (in blue) and the upregulation of the phonological network (in red). […] The strength of the individual inhibition of left AG after cTBS (effect sizes for AG/sham cTBS received from the effect of interest at x,y,z= -42, -64, 25) predicted the upregulation of left SMG […]”.